# Cell-Type-Specific Signalling Networks Impacted by Prostate Epithelial-Stromal Intercellular Communication

**DOI:** 10.3390/cancers15030699

**Published:** 2023-01-23

**Authors:** Kimberley C. Clark, Elizabeth V. Nguyen, Birunthi Niranjan, Yunjian Wu, Terry C. C. Lim Kam Sian, Lisa G. Horvath, Renea A. Taylor, Roger J. Daly

**Affiliations:** 1Cancer Program, Biomedicine Discovery Institute, Monash University, Melbourne, VIC 3800, Australia; 2Department of Biochemistry and Molecular Biology, Monash University, Melbourne, VIC 3800, Australia; 3Department of Anatomy and Developmental Biology, Monash University, Melbourne, VIC 3800, Australia; 4Monash Proteomics and Metabolomics Facility, Department of Biochemistry and Molecular Biology, Monash University, Melbourne, VIC 3800, Australia; 5The Garvan Institute of Medical Research, Sydney, NSW 2010, Australia; 6Chris O’Brien Lifehouse, Medical Oncology, Sydney, NSW 2050, Australia; 7The Kinghorn Cancer Centre, Sydney, NSW 2010, Australia; 8Cancer Research Division, Peter MacCallum Cancer Centre, University of Melbourne, Melbourne, VIC 3000, Australia; 9Department of Physiology, Monash University, Melbourne, VIC 3800, Australia

**Keywords:** signalling, tumour microenvironment, prostate cancer, mass spectrometry, fibroblast

## Abstract

**Simple Summary:**

Prostate cancer is the most commonly diagnosed cancer in men and one of the leading causes of cancer-related death. Intercellular communication between prostate epithelial cells and cells within the tumour microenvironment (TME), particularly cancer-associated fibroblasts (CAFs), are known to contribute to prostate cancer development and progression. However, the signalling mechanisms by which these reciprocal interactions occur are unknown. The aim of our study was to investigate the cell-type-specific signalling networks initiated by intercellular communication between prostate epithelial cells and patient-derived CAFs when grown together. We identified significant differences in the proteomic profiles of both cell types in this experimental setting. Functional analysis revealed that one of the top upregulated epithelial proteins following co-culture, transglutaminase-2 (TGM2), promotes epithelial cell migration and proliferation under co-culture conditions. This study identifies novel signalling pathways involved in intercellular communication in prostate cancer that may be exploited to improve management of this malignancy.

**Abstract:**

Prostate cancer is the second most common cause of cancer death in males. A greater understanding of cell signalling events that occur within the prostate cancer tumour microenvironment (TME), for example, between cancer-associated fibroblasts (CAFs) and prostate epithelial or cancer cells, may identify novel biomarkers and more effective therapeutic strategies for this disease. To address this, we used cell-type-specific labelling with amino acid precursors (CTAP) to define cell-type-specific (phospho)proteomic changes that occur when prostate epithelial cells are co-cultured with normal patient-derived prostate fibroblasts (NPFs) versus matched CAFs. We report significant differences in the response of BPH-1 benign prostate epithelial cells to CAF versus NPF co-culture. Pathway analysis of proteomic changes identified significant upregulation of focal adhesion and cytoskeleton networks, and downregulation of metabolism pathways, in BPH-1 cells cultured with CAFs. In addition, co-cultured CAFs exhibited alterations in stress, DNA damage, and cytoskeletal networks. Functional validation of one of the top differentially-regulated proteins in BPH-1 cells upon CAF co-culture, transglutaminase-2 (TGM2), demonstrated that knockdown of this protein significantly reduced the proliferation and migration of prostate epithelial cells. Overall, this study provides novel insights into intercellular communication in the prostate cancer TME that may be exploited to improve patient management.

## 1. Introduction

Despite advances in treatment, prostate cancer remains a significant cause of morbidity and mortality worldwide, with over 350,000 deaths attributed to this disease each year [1]. While there have been recent advances in treatment options and outcomes for prostate cancer, many of these are also associated with significant side effects that negatively affect the quality of life for patients. Identifying patients with prostate cancer that is more likely to progress to advanced disease stages and uncovering more effective treatment options would greatly benefit patient management. Traditionally, most research has focused on malignant epithelial cells; however, tumour cells do not exist in isolation and are involved in many different signalling interactions with the cells and factors of the surrounding tumour microenvironment (TME), many of which remain incompletely characterized.

Reciprocal communication between cell types within the TME plays a significant role in supporting tumour progression. Amongst the different cell types that comprise the TME, the role of fibroblasts has received particular attention, as these cells are one of the main components of the TME [2,3]. Importantly in prostate cancer, preclinical studies have shown that the presence of an altered or ‘reactive’ stroma is a predictor of poor prognosis and that high expression of cancer-associated fibroblast (CAF) markers Fibroblast Activation Protein (FAP) and Alpha-Smooth Muscle Actin (α-SMA) predict biochemical recurrence of disease in prostate cancer patients [4,5]. Moreover, prostate CAFs exhibit contrasting biological properties to normal prostate fibroblasts (NPFs) and influence prostate epithelial cells differently [3,6]. Tissue recombination experiments performed by Olumi et al. first demonstrated that recombining CAFs and benign prostate epithelial cells could stimulate tumour growth and alter the histology of the epithelial population, whereas this effect was not observed with NPFs [6]. Taylor and colleagues developed these findings further and demonstrated that tissue recombination of CAFs with nonmalignant BPH-1 prostate epithelial cells results in the formation of large tumour masses with poorly differentiated epithelial cells, compared to the growth of BPH-1 with NPF where the mass of epithelial cells exhibits extensive differentiation [3]. Furthermore, in vitro co-culture of BPH-1 cells with CAFs produced a more elongated BPH-1 cell morphology and increased cell motility in random cell migration assays [3]. In contrast, BPH-1 cells cultured with NPFs did not exhibit these phenotypic changes [3]. These studies highlighted a critical role for fibroblasts in altering the phenotype of prostate epithelial cells; however, mechanistic insights into these functional changes following co-culture are lacking, and the signalling mechanisms underpinning these responses remain unknown. Recently, we demonstrated key changes in the protein expression profiles between paired prostate cancer patient-derived NPFs and CAFs, which revealed an increase in the collagen/LOXL2/DDR2 signalling network in CAFs that promoted prostate epithelial cell migration [7]. However, our initial studies were performed in isolated cell cultures and did not take into account the signalling pathways activated in fibroblasts or epithelial cells in response to bidirectional signalling of cells in co-culture.

Until recently, comprehensive characterization of intercellular signalling between different cell types in co-culture was technically challenging. Previous approaches necessitated the use of cell sorting to isolate the individual cell populations from co-cultures prior to analysis, since the methodologies employed could not discriminate between DNA, RNA, or protein from different human cell types in co-culture. However, this approach has significant limitations, as unstable molecular changes that require continuous cell–cell communication, such as activation of phosphorylation-dependent signalling pathways, are not detected. Critically, these issues are circumvented by the powerful approach termed cell-type-specific labelling with amino acid precursors (CTAP) [8], which allows the proteomes of two different cell types in co-culture to be readily distinguished and, hence, characterization of the dynamic interplay that occurs between cell types within the tumour microenvironment [9,10]. However, this technique has yet to be employed to explore the reciprocal interactions that occur between prostate cancer epithelial cells and cells within the TME, such as CAFs.

In this study, we utilised CTAP to characterize how co-culture impacts the respective (phospho)proteomes of BPH-1 prostate epithelial cells co-cultured with prostate NPFs or CAFs. Importantly, we report significant differences in the response of BPH-1 cells to CAF co-culture compared to co-culture with paired NPFs. Moreover, functional interrogation of proteins exhibiting marked changes in BPH-1 cells upon CAF co-culture demonstrates that knockdown of transglutaminase-2 (TGM2) in nonmalignant BPH-1 and RWPE-1 prostate epithelial cells alters cell proliferation and migration, respectively.

## 2. Materials and Methods

### 2.1. Isolation of Nonmalignant Prostate Fibroblasts (NPF) and Cancer-Associated Fibroblasts (CAF) from Primary Prostate Tissue

Human prostate specimens were obtained following radical prostatectomy with the following human research ethics committee approvals: Cabrini Institute (03-14-04-08), Epworth HealthCare (34306 and 53611) and Monash University (2004/145). BPH-1 cells (kindly provided by Dr. Simon W. Hayward, Vanderbilt University, USA) were maintained in RPMI 1640 (School of Biomedical Sciences, Media and Prep Services, Monash University) + 5% fetal bovine serum (FBS) at 37 °C, 5% CO_2_, with media changes every 2–3 days. CAF and NPF were directly isolated from patient tissue and validated using in vivo tissue recombination experiments whereby CAF, but not NPF, promoted tumorigenicity of initiated prostate epithelial cells [11]. Briefly, benign and tumour regions were identified and excised by a trained pathologist. Whole tissue was collagenase-digested to release cells into suspension and cultured in fibroblast media (RPMI 1640 (School of Biomedical Sciences, Media and Prep Services, Monash University)) supplemented with phenol red, 5% heat-inactivated HyClone fetal bovine serum (HI-FBS; GE Healthcare, Chicago, IL, USA), 1 nM testosterone (Sigma-Aldrich, St Louis, MO, USA), 10 ng/mL basic fibroblast growth factor (Peprotech; Lonza Bioscience, Sydney, NSW, Australia), and P/S) [11]. Cells were maintained at 37 °C in 5% CO_2_, 5% O_2_ atmosphere, with media changes every 2–3 days. Patient-matched CAF and NPF cell lines were established from cancer and benign tissue pieces, respectively, and verified via immunohistochemistry at passage 4 to show homogenous expression of the fibroblast markers vimentin and α-smooth muscle actin and the absence of epithelial cytokeratins [3]. Given that CAF and NPF are primary and patient-derived, early passage (passage 3–8) CAF and NPF were used in this study unless otherwise stated.

### 2.2. Transfection and Cell-Type Labelling with Amino Acid Precursors

BPH-1 and NPF/CAF cells were transduced using the pGIPZ lentiviral packaging system (Thermo Fisher Scientific, Waltham, MA, USA) and the Lipofectamine^TM^ reagent (Thermo Fisher Scientific, Waltham, MA, USA). Mycobacterium tuberculosis (DDCM.tub-KDEL) (P0A5M4) diaminopimelate decarboxylase (DDC) and Proteus mirabilis lysine racemase (LyrM37-KDEL) (M4GGR9) were synthesized by GeneArt [10]. BPH-1 cells were transduced with LyrM37-KDEL, while CAF/NPF cells were transduced with DDCM.tub-KDEL. Transduction was confirmed by Western blot analysis. All cells were grown in RPMI 1640 (deficient for L-lysine and L-arginine) Cat#R8999-03A (US Biological, Salem, MA, USA) supplemented with 5% (*v*/*v*) dialyzed FBS (Invitrogen, Waltham, MA, USA) and 0.3 mM L-arginine (A8094, Sigma-Aldrich, St. Louis, MO, USA). BPH-1+LyrM37-KDEL cells were grown with 2.5 mM “heavy” isotope variant D-lysine-3,3,4,4,5,5,6,6-d8 2HCl (Cat# D-6367, CDN isotopes, Quebec, Canada). CAF/NPF+DDCM.tub-KDEL cells were grown with 5 mM “light” diaminopimelate (DAP) for 5 days. Herein, “heavy” refers to cells that are isotopically labelled by utilizing heavy D-lysine as an amino acid source, which incorporates stable-isotope-labelled lysine into the proteome. Conversely, “light” refers to cells that utilize DAP and remain unlabelled. Concomitantly to investigate reciprocal signalling in co-culture, “heavy” BPH-1+LyrM37-KDEL cells were co-cultured in direct contact with “light” CAF-/or NPF+DDCM.tub-KDEL cells in a 10:1 seed ratio. Co-culture media had only 0.5% dialysed FBS and was maintained at 37 °C in 5% CO_2_ and 5% O_2_ atmosphere. After 5 days, each monoculture and co-culture condition were washed 2× with 1 × TBS, lysed with 4% sodium deoxycholate (SDC) (*w*/*v*) in 100 mM TEAB (pH 8–8.5), and immediately heated for 5 min at 95 °C to facilitate lysis and inactivate endogenous proteases and phosphatases. Lysates were then sonicated and centrifuged at 16,000× *g* for 10 min at 4 °C to clear cell debris, and protein concentration was determined by BCA. One hundred micrograms of each sample were reduced with 5 mM Tris (2-carboxyethyl)phosphine (TCEP) at 55 °C for 30 min then subjected to alkylation with 20 mM Chloroacetamide in the dark for 30 min and digestion with Lys-C and trypsin (1:100) O/N at 37 °C with shaking (1500 rpm) overnight. Tryptic digests were acidified with 10% TFA to pH 2–3 and centrifuged at 10,000× *g* for 10 min at 4 °C. SDC was removed, and supernatant was isobaric tandem mass tag (TMT, Thermo Fisher Scientific, Waltham, MA, USA) labelled before being combined and desalted with a C18 column (Thermo Fisher Scientific, Waltham, MA, USA) and eluted with 0.1% trifluoroacetic acid (TFA)/40% acetonitrile (ACN). Peptides were dried with a SpeedVac (Thermo Fisher Scientific, Waltham, MA, USA) and resuspended in 2% ACN/0.1% formic acid (FA) prior to mass spectrometry (MS) analysis.

### 2.3. Phosphopeptide Enrichment

Following TMT labelling and combining all samples, phosphopeptides were enriched as previously described [12]. Briefly, peptides were enriched with a 12:1 TiO_2_ bead (Cat No. 5010-21315, GL Sciences, Tokyo, Japan) to protein ratio for 5 min at 40 °C with shaking (2000 rpm). Phosphopeptides were eluted with EP elution buffer (5% NH4OH in 40% *v*/*v* ACN) prior to consequent desalting with an in-house-prepared SDB-RPS (Empore^TM^, CDS Analytical, Oxford, PA, USA) stage tip and eluted with 20 μL of 25% NH4OH in 60% *v*/*v* ACN and evaporated to dryness in a SpeedVac. The dried peptides were reconstituted in 2% (*v*/*v*) ACN/0.3% (*v*/*v*) TFA.

### 2.4. Mass Spectrometry Analysis

Samples were analysed on an UltiMate 3000 RSLC nano LC system (Thermo Fisher Scientific, Waltham, MA, USA). Peptides were loaded via an Acclaim PepMap 100 trap column (100 μm × 2 cm, nanoViper, C18, 5 μm, 100 Å, Thermo Fisher Scientific, Waltham, MA, USA). One μg of peptides as measured by a nanodrop 1000 spectrophotometer (Thermo Fisher Scientific, Waltham, MA, USA) was loaded on the precolumn with microliter pickup. Peptides were eluted using a 2 h linear gradient of 80% (*v*/*v*) ACN/0.1% FA flowing at 250 nL/min using a mobile phase gradient of 2.5–42.5% ACN. The eluting peptides were interrogated using synchronous precursor selection tandem MS/MS/MS (SPS-MS3) workflow to eliminate isolation interference and dynamic range compression, which are commonly observed in isobaric tag-based quantitative proteomics experiments. For peptide identification, raw data files were analysed with the MaxQuant Version 1.6.0.16 analysis software using default settings. Group-specific parameters were set to Type Reporter ion MS3 and TMT labels loaded under isobaric labels. Enzyme specificity was set to Trypsin/P and LysC/P, minimal peptide length of 6, and up to 4 missed cleavages were allowed. Search criteria included carbamidomethylation of cysteine as a fixed modification; oxidation methionine; acetyl (protein N terminus); deamidation (NQ); custom “heavy” deuterated lysine-d8 modification (+8.05021396); and phosphorylation of serine, threonine, and tyrosine as variable modifications. The mass tolerance for the precursors was 4.5 ppm and for the fragment ions was 20 ppm. The DDA files were searched against the human UniProt fasta database (v2015-08, 20, 210 entries concatenated with DDCM.tub-KDEL and LyrM37-KDEL peptides). Only lysine containing peptides was included in the CTAP quantitative analysis.

### 2.5. Mass Spectrometry Statistical Analysis

Peptide intensities were log_2_ transformed, imputation via normal distribution with Perseus software before quantile normalization. A FC of co-culture/monoculture of over 2 in either direction and a *p*-value < 0.05 were the criteria for differential expression. Differential proteins observed across all three replicates were further examined.

### 2.6. Functional Annotation Analysis

Functional annotation of the (phospho)proteome was conducted using InnateDB [13]. Over-represented functional categories among proteins enriched in each mono/co-culture were evaluated using hypergeometric distribution with Benjamini Hochberg corrected *p*-value. Criteria for reported functional enrichment required an adjusted *p*-value < 0.05 and >2 proteins mapping to a functional pathway. Whole proteome and phosphosites were assessed separately for enrichment of functional pathways. Pathways common between the two were represented visually with STRING [14] and Cytoscape [15].

### 2.7. Western Blot Analysis

Standard Western blots were undertaken using RIPA (radioimmunoprecipitation assay) lysates as previously described [16]. A total of 10–20 ug of lysate was resolved via SDS-PAGE. Primary antibodies were as follows: anti-vinculin (1:5000) (Cat#4650, Cell Signalling Technology, Danvers, MA, USA), mouse anti-β-actin (1:5000) (Cat#sc-69879, Santa Cruz Biotechnology, Dallas, TX, USA), rabbit anti-TGM2 (1:1000) (Cat#ab2386, Abcam, Cambridge, UK), rabbit anti-FTL (Cat#ab69090, Abcam, Cambridge, UK), rabbit anti-ANXA6 (Cat#31026, Abcam, Cambridge, UK), mouse anti-MVP (Cat#ab97311, Abcam, Cambridge, UK), rabbit anti-NDRG1 (Cat#5196S, Cell Signalling Technology, Danvers, MA, USA), and rabbit anti-SOD2 (Cat#13194S, Cell Signalling Technology, Danvers, MA, USA). HRP-conjugated goat anti-mouse (Cat#1706516, Bio-Rad, Hercules, CA, USA) and goat anti-rabbit (Cat#1706515, Bio-Rad, Hercules, CA, USA) secondary antibodies were used at 1:5000 and 1:3000, respectively.

### 2.8. siRNA Knockdown

BPH-1 and RWPE-1 (American Type Culture Collection; ATCC) benign prostate epithelial cell lines were used in assays where the proteins identified by mass spectrometry analysis as differentially expressed were transiently knocked down using siRNA. Gene knockdown was performed using Dharmacon OnTargetPlus siRNA targeting pools or siOTP nontargeting control SMARTPool, as per the manufacturer’s instruction with DharmaFECT transfection reagent 1. Protein expression was confirmed by Western blot, as previously described.

### 2.9. MTS Proliferation Assay for Cells in Monoculture

Analysis of the proliferation of BPH-1 prostate epithelial cells was performed by MTS assay. BPH-1 cells were seeded in a 96-well plate at 5 × 10^3^ cells per well. At 0, 24, 48, and 72 h timepoints, 10 µL of MTS reagent (CellTiter 96(R) Aqueous One Solution MTS assay reagent, Promega, WI, USA) was added to the culture and incubated for 30 min. Absorbance was measured using a CLARIOstar plate reader (BMG Labtech, Mornington, VIC, Australia) as a measure of cell number.

### 2.10. Scratch Assay

Analysis of the migration of BPH-1 prostate epithelial cells was performed by scratch assay over 16 h. BPH-1 cells were seeded into a 24-well plate at 1 × 10^5^ cells per well and allowed to adhere overnight. Cells were treated with 10 µg/mL mitomycin-c for 2 h. Cells were washed and replaced with complete culture medium, then 3 h later a linear scratch was made in the centre of each well using a P200 pipette tip. Cells were imaged using a Leica DMi8 microscope (Leica Microsystems, Wetzlar, Germany) with 3 fields of view imaged for each well at 10× magnification every 30 min for 16 h. The width of the scratch was assessed using the FIJI open-source software measurement tool [17].

### 2.11. Co-Culture Proliferation Assay

At the time of initiating siRNA knockdown in BPH-1 GFP+ cells, CAFs were seeded at 5 × 10^3^ cells per well in a 48-well plate. At 48 h post-addition of siRNA to BPH-1 cultures, BPH-1 cells were trypsinized and seeded directly onto the fibroblast cells at 5 × 10^3^ cells per well as a contact co-culture. GFP fluorescent images were taken on a Leica DMi8 microscope with 4 fields of view images for each well at 4× magnification at day 0 and day 3 timepoints. Images were analysed using FIJI open-source software to measure the GFP % threshold.

### 2.12. Co-Culture Random Cell Migration Assay

As described for the co-culture proliferation assay, CAFs were seeded in a 48 well plate. At 48 h post-addition of siRNA to BPH-1 cultures, epithelial cells were treated with 10 µg/mL mitomycin-c for 2 h. Cells were washed and incubated with complete culture medium for a further 2 h. Cells were trypsin-treated and seeded onto the CAFs. Cells were allowed to settle into culture for 4 h. BPH-1 cells were imaged using a Leica DMi8 microscope (Leica Microsystems, Wetzlar, Germany) with 3 fields of view imaged for each well at 4× magnification every 30 min for 16 h. Cell movement was analysed using ImageJ software to track distance and velocity of GFP+ cells.

## 3. Results

### 3.1. Global Proteomic Analysis of NPF or CAF Co-Cultures with BPH-1 Prostate Epithelial Cells Using CTAP

Recent advances in “omics” technologies have overcome the need to isolate cell types for downstream analysis, which may compromise detection of labile modifications such as protein phosphorylation and allow for different cell types in continuous cellular communication to be analysed. Such approaches can be employed to resolve intercellular signalling pathways within the prostate cancer TME for further preclinical investigation and research translation. In this study, we utilized CTAP to investigate intercellular signalling pathways between nonmalignant BPH-1 prostate epithelial cells and paired patient-derived NPF and CAFs (Figure 1A). CTAP was combined with isobaric tandem mass tag (TMT) labelling in order to achieve this. Briefly, BPH-1 cells and CAF/NPF cells were programmed to express LyrM37-KDEL and DDCM.tub-KDEL, respectively, with expression confirmed by Western blot analysis (Figure 1B). Differential isotopic labelling of cells was achieved with BPH-1+-LyrM37-KDEL grown in the presence of heavy deuterium D-lysine(-d8), while CAF/NPF+-DDC-M.tub-KDEL were grown in diaminopimelate (DAP)-supplemented medium (Figure 1A).

To characterize the influence of fibroblast co-culture on the epithelial cell proteome of BPH-1 cells, quantitative analysis was performed to identify proteins differentially expressed in co-cultures versus monocultures. Upon co-culture with CAFs, there were 13 BPH-1 proteins that significantly increased in abundance greater than twofold-change upon co-culture with CAFs, while there were 4 BPH-1 proteins that consistently decreased greater than twofold-change (Figure 2; quadrant IV) (Table 1). With NPFs, nine proteins were upregulated and four downregulated (Figure 2; quadrant I). There were five CAF proteins that were increased upon co-culture with BPH-1 and 11 proteins that decreased (Figure 2; quadrant III). Comparatively, there was a total of seven NPF proteins that were increased in abundance following BPH-1 co-culture (Figure 2; quadrant II). There were six NPF proteins downregulated following BPH-1 co-culture, three of which were also downregulated in CAFs (Figure 2; quadrant II). This analysis demonstrated a large degree of overlap in the differentially regulated BPH-1 proteins in NPF or CAF co-culture. For example, 46% of the BPH-1 proteins significantly upregulated following CAF co-culture were also observed to be upregulated following NPF co-culture, including proteins such as TGFB1, VIM, and TGM2.

The CTAP analysis allowed a direct comparison of the influence of CAF- and NPF-co-culture on the BPH-1 proteome. Of the 12 BPH-1 proteins significantly upregulated in CAF co-culture, six proteins were only significantly increased in CAF co-culture, not with NPFs: STOM, NDRG1, FN1, FAM114A1, LGALS1, and CTSB (Figure 3A, Table 1). While a large degree of overlap was observed between BPH-1 proteins upregulated in CAF and NPF co-culture, several of the top upregulated proteins exhibited striking fold-change increases that were considered worthy of further functional exploration, such as TGM2 (36- and 24-fold in CAF and NPF co-culture, respectively). Of the four BPH-1 proteins significantly decreased in CAF co-culture (ACAT2, PSME2, SCD, and GLS), only PSME2 was also significantly decreased with NPFs (Figure 3B, Table 1).

In addition to changes in BPH-1 proteins, some proteins were differentially expressed in CAFs and NPFs following BPH-1 co-culture. Of the five CAF proteins with increased protein abundance in co-culture, two were increased in CAFs but not NPFs: PRKDC and GBA (Figure 3C, Table 2). Of the 11 CAF proteins that showed a decrease in abundance in BPH-1 co-culture, seven were only significantly reduced in CAFs, not NPFs, when they were co-cultured: TAGLN, RAB5A, MGST1, MARCKS, CNN3, CALD1, and ITGAV (Figure 3D, Table 2).

Overall, these data demonstrate that there is a significant overlap in protein expression changes in BPH-1 cells co-cultured with CAFs or NPFs, and in the co-cultured fibroblasts, but CAF-specific changes can be detected, highlighting the importance of co-culture approaches and application of CTAP.

### 3.2. Global Phosphoproteomic Analysis of NPF or CAF Co-Cultures with BPH-1 Prostate Epithelial Cells Using CTAP

Quantitative analysis of phosphoproteins between cell conditions was performed to characterize changes in cellular signalling events that may not be captured by changes in total protein abundance. A TiO_2_-enrichment workflow was used to identify 10 BPH-1 phosphosites that significantly increased in abundance upon co-culture with CAFs (Figure 4; Quadrant IV), while five phosphosites significantly decreased upon co-culture (Figure 4; Quadrant IV) (Appendix A). Furthermore, five CAF phosphosites increased in abundance upon co-culture with BPH-1 and four decreased (Figure 4; Quadrant III) (Appendix A). Of particular interest, the serine 902 phosphorylated form of SYNPO2 was increased in BPH-1 cells by ~40-fold in CAF co-culture, but this was not observed upon NPF co-culture (Figure 4). Similar to the observations made in the whole proteome CTAP analysis, a large degree of overlap of differentially regulated BPH-1 phosphosites was observed between CAF and NPF co-culture, with 44% of the phosphosites significantly upregulated in BPH-1 cells following CAF co-culture also upregulated following NPF co-culture, including VIM S430 and NEXN S80 (Figure 4; Quadrant I and IV). Interestingly, for vimentin (VIM), an increase in whole protein abundance (>10-fold) and phosphorylation at S430 (~fourfold) was observed for BPH-1 cells in either NPF or CAF co-culture (Table 1 and Appendix A). This indicates that while vimentin protein abundance increases, relative phosphorylation (i.e., normalized for total protein) actually decreases.

### 3.3. Signalling between BPH-1 Epithelial Cells and CAFs in Co-Culture

In order to characterize pathways and networks that were altered in BPH-1 cells in response to co-culture with CAFs, we undertook STRING/Cytoscape network analysis of BPH-1 proteins and phosphosites significantly altered in abundance in co-culture. Pathway analysis was undertaken separately for changes in whole proteome and phosphosites, while common pathways were combined for network analysis. Given the relatively small number of proteins identified as differentially regulated, a more relaxed 1.2-fold-change cut-off was used for this analysis (Table 1 and Appendix A). Of note, some of these protein/sites were also modulated upon NPF co-culture. This identified the focal adhesion and cytoskeleton pathways as enriched in BPH-1 cells in CAF co-culture compared to monoculture (Figure 5A), with an interaction hub of actin-binding proteins, including SYNPO2 and NEXN, associated with both of these pathways. SYNPO2 is of particular interest as it has previously been associated with enhanced stress-fibre assembly and the formation of immature focal adhesions in the PC-3 prostate cancer cell line [18]. TGM2, the most markedly upregulated BPH-1 protein in CAF-co-culture (Figure 3A), is also associated with focal adhesion signalling. Network analysis of proteins/phosphosites that were downregulated revealed an association with “metabolism” (e.g., ACLY, SCD, ACAT2, and GLS) (Figure 5B).

STRING/Cytoscape network analysis of CAF proteins/phosphosites with increased abundance in co-culture with BPH-1 cells demonstrated an enrichment in pathways for “cellular response to stress” and “DNA damage/telomere stress-induced senescence” (Figure 6A). Two separate hubs of ribosomal (RPS11, RPS4X, and RPL12) and histone proteins (HIST1H2BM, HIST1H1B, and HIST1H2AJ) were linked to the upregulation of stress pathway signalling in CAFs following co-culture. The histone interaction hub was also implicated in DNA damage signalling, likely due to the role of histones in chromatin remodelling for the DNA damage response. Analysis of CAF proteins with decreased abundance in BPH-1 co-culture highlighted an association with “cytoskeleton” signalling (Figure 6B). The proteins associated with decreased cytoskeleton signalling in CAFs following BPH-1 co-culture (Figure 6B) largely differed from those identified as upregulated in BPH-CAF co-culture (Figure 5A).

Analysis of altered proteins in co-culture demonstrated reciprocal rewiring of signalling networks when prostate epithelial cells are co-cultured with fibroblasts. Well-established pathways frequently associated with functional changes in cell growth and migration such as “focal adhesion” and “cytoskeleton” were upregulated in BPH-1 epithelial cells in CAF co-culture and warranted further investigation through functional validation experiments.

### 3.4. Functional Validation

To further understand the functional importance of the pathways identified in this study, we assessed the functional role of five CAF-induced BPH-1 proteins that were significantly upregulated following co-culture: TGM2, FTL, ANXA6, NDRG1, and MVP. In addition, SOD2 was included in functional validation due to its recently reported roles in prostate cancer metabolism and upregulation in other malignancies, while also demonstrating a trend toward increased expression in BPH-1 cells in CAF co-culture [19,20]. Given that some targets were upregulated more than 30-fold, which may be hard to reliably reproduce when overexpressing these proteins, an siRNA knockdown approach was instead used in prostate epithelial cells (BPH-1 and RWPE-1) (Figure 7A). This allowed us to determine whether these proteins represent valid therapeutic targets for future investigation.

Individual knockdown of these six proteins in BPH-1 cells did not significantly affect cell proliferation in monoculture (Figure 7B). Additionally, knockdown of these target proteins in a second independent immortalized benign prostate epithelial line, RWPE-1, also showed no difference in proliferation in monoculture (Figure 7B). However, given that our study demonstrated reciprocal rewiring of signalling networks when prostate epithelial cells are co-cultured with fibroblasts, it was important to repeat the assay under co-culture conditions. To this end, BPH-1 or RWPE-1 cells with siRNA knockdown of the proteins investigated above were cultured with CAF332R fibroblasts. Importantly, a 30% decrease in cell number was observed in BPH-1 cells with knockdown of TGM2 under these assay conditions (Figure 7C). Additionally, BPH-1 cells with ANXA6 and NDRG1 knockdown also showed a decrease in cell proliferation in CAF co-culture (Figure 7C). However, these changes were not observed when RWPE-1 cells were used in the CAF co-cultures.

Given the identification of “focal adhesion” and “cytoskeleton” pathways in upregulated BPH-1 proteins in CAF co-culture, it was important to investigate the role of the selected candidates in cell migration. Similar to proliferation, there was no significant difference in the migratory ability of BPH-1 or RWPE-1 cells in a scratch closure assay upon candidate knockdown (Figure 8A). However, upon co-culture with CAFs, RWPE-1 cells with TGM2 knockdown demonstrated a reduction in random cell migration, although this was not observed in BPH-1 cells (Figure 8B). Consequently, these data indicate that TGM2 can also impact CAF-dependent cell migration in co-cultured prostate epithelial cells, supported by the pathway analysis that identified TGM2 as associated with the “focal adhesion” signalling network upregulated following CAF co-culture. Moreover, a number of known TGM2 network partners, including TGFB, FN1, and particular cytoskeletal regulators [21,22], are also upregulated at the protein or phosphosite level in BPH-1 cells following CAF co-culture (Appendix A). Overall, these data support important functional roles for CAF-regulated TGM2, ANXA6, and NDRG1 in migration and/or proliferation of co-cultured prostate epithelial cells, but there is evidence of heterogeneity in the biological response.

## 4. Discussion

In this study, CTAP was employed to analyse intercellular signalling pathways between BPH-1 cells and CAFs to identify proteins that are altered in co-culture in response to cell–cell interactions. This approach allows the identification of alterations in signalling pathways that are likely to be involved in tumorigenesis. Pathway analysis of proteomic changes identified significant upregulation of focal adhesion and cytoskeleton networks, and downregulation of metabolism pathways, in BPH-1 cultures with CAFs. In addition, co-cultured CAFs exhibited alterations in stress, DNA damage, and cytoskeletal networks.

Herein, we identified transglutaminase-2 (TGM2) as a top regulated protein increased in BPH-1 cells in BPH-1-CAF co-culture, with a striking 36-fold increase upon CAF co-culture. TGM2 is a multifunctional enzyme with transglutaminase crosslinking, G protein signalling, and kinase activities that are postulated to play a role in many disease states, including celiac disease and cancer [23]. TGM2 is implicated in a number of cancers, including glioblastoma, endometrial cancer, and renal clear cell carcinoma [24,25,26,27,28]. TGM2 expression is increased in renal clear cell carcinoma compared to normal tissue and forms part of an ECM signature that is significantly associated with overall patient survival [27]. Moreover, TGM2 represents a potential therapeutic target, as TGM2 inhibition promotes cell death and chemosensitivity in in vivo models of glioblastoma [24]. However, a role for TGM2 in prostate cancer development and progression has not been demonstrated. This study indicates that TGM2-knockdown in benign prostate epithelial cells can reduce cell growth and migration in a CAF-dependent manner, warranting further investigation of TGM2′s regulation and role in prostate cancer initiation and progression. In this context, it will be interesting to determine whether the regulation of TGM2 (and other identified targets) is at the protein or mRNA level, which will require application of transcriptomic approaches that provide spatial information, such as digital spatial profiling. In addition, it is currently unclear as to whether TGM2 regulation in co-culture is mediated by direct cell–cell contact, the cell matrix, or secreted soluble factors.

Annexin A6 (ANXA6) and N-myc-downregulated gene 1 (NDRG1) were also upregulated in BPH-1 cells following CAF co-culture. Functional analysis of BPH-1 cells with siRNA knockdown of these targets indicates that they play a role in promoting CAF-dependent epithelial cell proliferation. ANXA6 is a member of a highly conserved family of calcium-dependent membrane and phospholipid-binding proteins [29] and is involved in a range of cell signalling pathways dependent on intracellular calcium levels [30]. In conditions with increased intracellular calcium, ANXA6 translocates to the plasma membrane and endosomal compartments where it primarily acts as a multifunctional scaffold protein, recruiting signalling proteins and influencing actin dynamics [30,31]. Given the array of functional roles for ANXA6, its role in carcinogenesis is also varied. Increased expression of ANXA6 is associated with more advanced disease stages in cervical cancer and pancreatic cancer [32,33]. In particular, exosome-derived ANXA6 from breast cancer stem cells and pancreatic-cancer-associated fibroblasts promotes more aggressive tumour growth in vivo [33,34]. Conversely, ANXA6 expression is negatively correlated with melanoma, gastric cancer, and chronic myeloid leukaemia disease progression [35]. A role for ANXA6 in prostate cancer is unclear. A study by Xin et al. reported a decrease in ANXA6 gene expression in more malignant disease stages compared to benign prostate hyperplasia following a meta-analysis of four gene profiling studies [36]. However, the role of the prostate TME in influencing ANXA6 expression remains unknown. The data presented here suggest a potential role for CAFs to upregulate ANXA6 expression in early disease stages in prostate cancer, which promotes increased epithelial cell proliferation, suggesting the possibility of therapeutic targeting of this CAF-specific signalling mechanism.

NDRG1 is widely known as an oncogenic signalling disruptor involved in numerous signalling pathways in oncogenesis, through mechanisms which largely remain unclear. A number of studies in prostate cancer have demonstrated that NDRG1 associates with androgen receptors and inhibits a range of critical signalling pathways, including EGFR, HER2, HER3, PI3K, and STAT3 [37,38,39,40]. Furthermore, a study by Lim et al. demonstrated that there is a negative correlation between NDRG1 expression and prostate specific antigen (PSA) in prostatectomy patients that go on to develop metastatic disease [37]. Downregulation of NDRG1 in C4-2 and PC-3 prostate cancer cell lines promotes an increase in cell invasion and migration in cell culture, and suppression of AR/NDRG1 signalling has been suggested as a potential mechanism that promotes castrate-resistant prostate cancer (CRPC) progression [38]. Conversely, a recent comparative study by Lage-Vickers et al. highlighted that NDRG1 expression is significantly increased in prostate tumour mRNA compared to normal adjacent tissue [41]. Moreover, high NDRG1 expression is associated with worse overall survival in treatment-naïve prostate cancer patients [41]. A protumorigenic role for NDRG1 is also supported in lung and rectal cancer, where increased NDRG1 is linked to reduced response to cisplatin and radiation therapy, respectively [42,43]. Additionally, inflammatory breast cancer patients with high NDRG1 expression exhibit reduced overall survival vs. patients with low NDRG1 expression; likewise, more aggressive disease was observed in NDRG1-high mouse models of inflammatory breast cancer [44,45]. In the study presented here, BPH-1 cells in co-culture with CAFs exhibited an increase in NDRG1 protein expression relative to both BPH-1 monoculture and NPF-co-culture, supporting a CAF-mediated role for NDRG1 signalling in early disease stages of prostate cancer.

More broadly, this study demonstrates changes in critical signalling pathways in both premalignant prostate epithelial cells and CAFs when in co-culture. Pathway analysis conducted on BPH-1 proteins significantly increased upon co-culture with CAFs showed an association with “focal adhesion” and “cytoskeleton”. The role of these pathways is widely established in contributing to cell motility and migration. This is consistent with Clark et al., where functional changes in prostate epithelial cell migration were observed following CAF co-culture, and a separate study by our team that detailed increased prostate epithelial cell migration in response to an upregulated CAF protein, LOXL2 [7]. In an array of cancers, including prostate cancer, increased cell migration is associated with metastasis and subsequently more advanced stages of disease. The current study highlights the importance of CAFs in contributing to this process in early/premalignant disease stages as modelled by the benign BPH-1 cells. Our findings strengthen the rationale for exploring CAF-targeted chemopreventative strategies directed against prostate cancer development.

Interestingly, pathway analysis of CAF proteins significantly dysregulated upon BPH-1 co-culture showed an association with “cellular response to stress”. The role of cellular stress in cancer is an area of increased research interest as cancer cells display increases in energy-demanding processes, such as protein synthesis, during tumorigenesis that must be delicately balanced by adaptive stress responses for sustained cell growth and survival [46,47]. In prostate cancer, therapeutically targeting this stress response via inhibition of phosphorylated eukaryotic initiation factor 2 α (P-eIF2α) promoted a cytotoxic response in metastatic prostate PDXs [48]. Based on our findings, further investigation regarding how increased cellular stress in CAFs participates in prostate tumour development is required. Importantly, a recent study highlighted the critical role of cellular stress signalling within the TME and the impact this has on tumour development in preclinical models of melanoma and pancreatic cancer [49]. Verginadis et al. demonstrated that activating transcription factor 4 (ATF4) plays an important role in CAF activation and subsequent tumour vascularization and growth [49]. ATF4 is a stress-responsive gene that is upregulated in response to activation of integrated stress response signalling via eIF2α phosphorylation. Host mice with ATF4 knockout presented with significant delays in melanoma tumour growth following tumour cell inoculation compared to wildtype hosts. Single-cell RNAseq and clustering analysis of these tumours uncovered a significant downregulation of CAF markers, such as *Acta2* and *Pdgfrb*, in the CAF component of ATF4 knockout mice [49]. Future studies should investigate the contribution of CAFs to stress signalling in prostate cancer and how this may inform novel strategies for prostate cancer disease management.

## 5. Conclusions

Despite significant advances in our understanding of the contrasting properties between CAFs and NPFs in prostate cancer, the intercellular signalling between prostate epithelial cells and CAFs is still poorly understood, and it is likely that considerable untapped potential remains within the prostate cancer TME for biomarker and therapeutic development. This study characterizes the effects of intercellular communication between prostate epithelial cells and prostate cancer CAFs using the powerful approach of CTAP to distinguish between the global whole and phosphoproteomic profiles of different cell types upon sustained co-culture. We demonstrated that proteins identified by this approach significantly contribute to prostate epithelial cell growth and migration in culture and represent potential therapeutic targets and/or biomarkers warranting further preclinical validation.

## Figures and Tables

**Figure 1 cancers-15-00699-f001:**
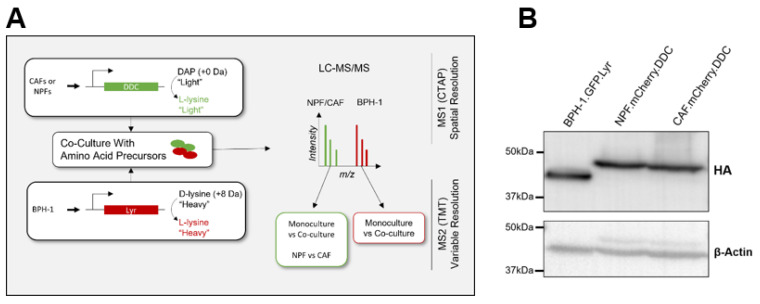
Application of CTAP to benign prostate epithelial cell–prostate fibroblast intercellular communication. (**A**) CTAP provides spatial resolution, identifying which cell type the MS-detected peptides are derived from. Patient-derived fibroblasts (NPFs and CAFs) and BPH-1 cells were modified to express DDCM.tub-KDEL (DDC) and LyrM37-KDEL (Lyr), respectively. Cells were cultured with differentially isotopically-labelled amino acid precursors to allow identification of the cell-type of origin during LC-MS/MS proteomic analysis. Discrimination between additional variables was achieved by differential isobaric tandem mass tag (TMT) labelling of peptides. Adapted from Tape et al., 2014 [10]. (**B**) Western blot validation that the BPH-1 and NPF/CAFs express HA-tagged Lyr and DDC, respectively.

**Figure 2 cancers-15-00699-f002:**
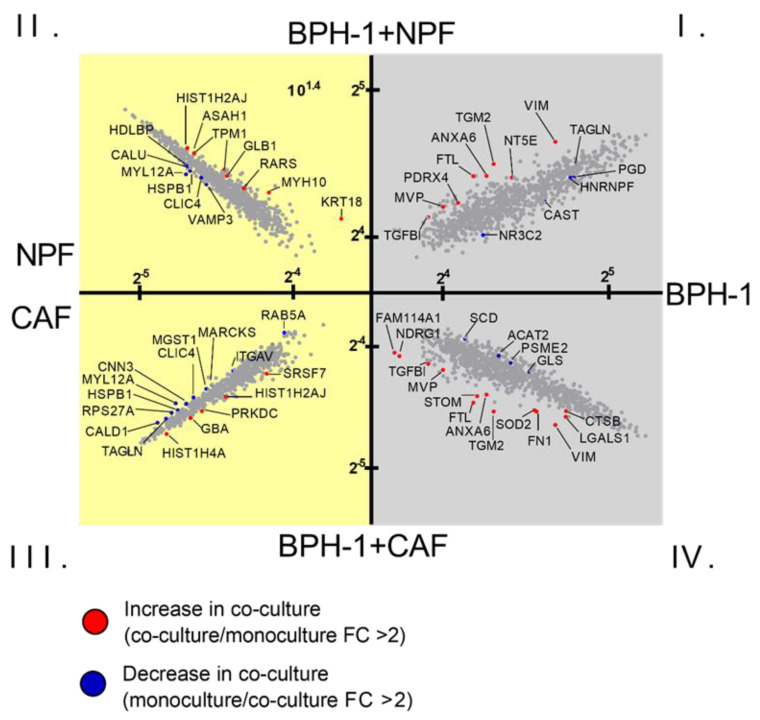
X-plot of differentially abundant proteins in BPH-1/fibroblast co-culture compared to monoculture. Scatter plots representing mean log2-transformed protein abundance; *x*-axis represents abundance in monoculture, and *y*-axis represents abundance in co-culture. Proteins significantly upregulated (red dots; protein level in co-culture/corresponding monoculture) and downregulated proteins (blue dots; protein level in monoculture/corresponding co-culture) in co-culture were selected using a cut-off value of FC > 2. Proteins closer to the *y*-axis represent an increase of expression, while proteins closer to the *x*-axis represent a decrease of expression upon co-culture. Grey background: BPH-1 protein expression, yellow background: fibroblast protein expression. FC: fold change. Quadrant I: BPH-1 proteins in monoculture compared to BPH-1 proteins in co-culture with NPFs, Quadrant II: NPF proteins in monoculture compared to NPF proteins in co-culture with BPH-1 cells, Quadrant III: CAF proteins in monoculture compared to CAF proteins in co-culture with BPH-1 cells, Quadrant IV: BPH-1 proteins in monoculture compared to BPH-1 proteins in co-culture with CAFs. Data represent mean protein abundance across 3 biological replicates with NPFs and CAFs from a single patient.

**Figure 3 cancers-15-00699-f003:**
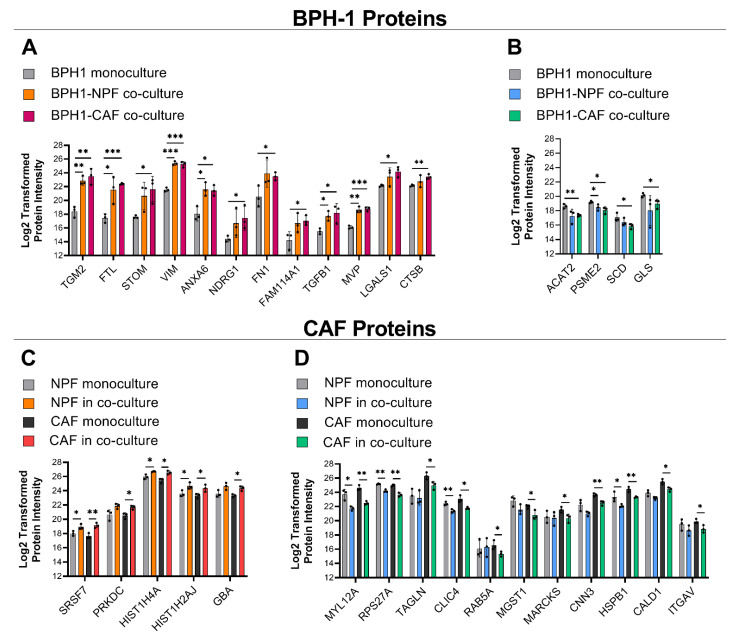
BPH-1 and fibroblast proteins differentially expressed in co-culture. A summary of BPH-1 proteins with (**A**) increased and (**B**) decreased expression following co-culture with CAFs, and comparison with effect of NPF co-culture, and a summary of CAF proteins with (**C**) increased and (**D**) decreased expression following co-culture with BPH-1 cells, and a comparison with effects in NPFs when they are co-cultured with BPH-1 cells. Only proteins with a >twofold change in either direction in CAF co-culture conditions presented. Data represent mean protein abundance across 3 biological replicates with NPFs and CAFs from a single patient. Mean ± SD, *n* = 3. Student’s *t*-test. * *p* < 0.05, ** *p* <0.01, ****p* < 0.001.

**Figure 4 cancers-15-00699-f004:**
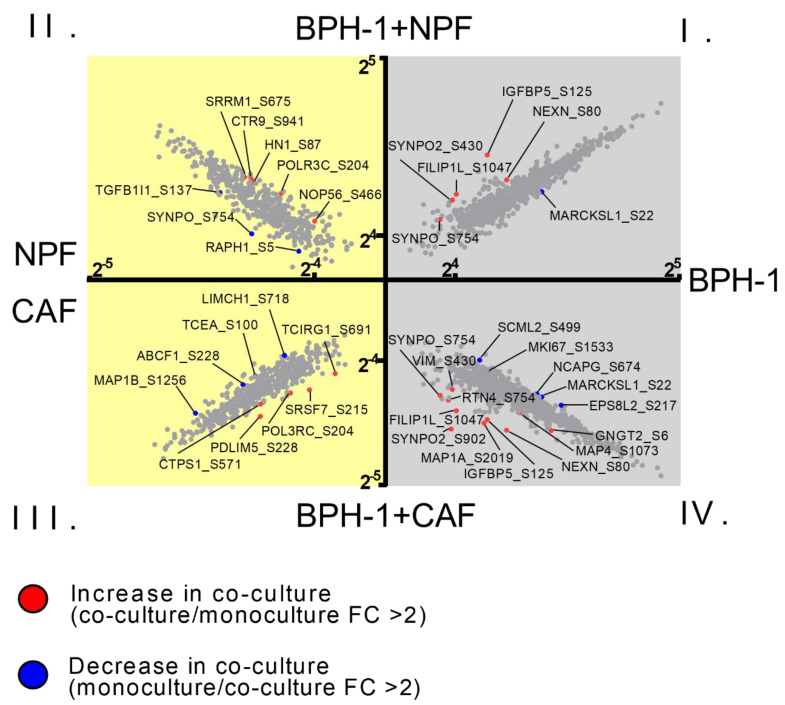
X-plot of differentially abundant phosphosites in BPH-1/fibroblast co-culture compared to monoculture. Scatter plots representing mean log2-transformed phosphosite abundance; *x*-axis represents abundance in monoculture, and *y*-axis represents abundance in co-culture. Significantly upregulated phosphosites (red dots; phosphosite level in co-culture/corresponding monoculture) and downregulated phosphosites (blue dots; phosphosite level in monoculture/corresponding co-culture) in co-culture were selected using a cut-off value of FC > 2. Phosphosites closer to the *y*-axis (red) represent a significant increase of expression, while phosphosites closer to the *x*-axis (blue) represent a decrease of expression upon co-culture. Grey background: BPH-1 phosphosite levels, yellow background: fibroblast phosphosite levels. Quadrant I: BPH-1 phosphoproteins in monoculture compared to BPH-1 phosphoproteins in co-culture with NPFs, Quadrant II: NPF phosphoproteins in monoculture compared to NPF phosphoproteins in co-culture with BPH-1 cells, Quadrant III: CAF phosphoproteins in monoculture compared to CAF proteins in co-culture with BPH-1 cells, Quadrant IV: BPH-1 phosphoproteins in monoculture compared to BPH-1 phosphoproteins in co-culture with CAFs. FC, fold change. Data represent phosphosite abundance across 3 biological replicates from NPFs and CAFs from a single patient.

**Figure 5 cancers-15-00699-f005:**
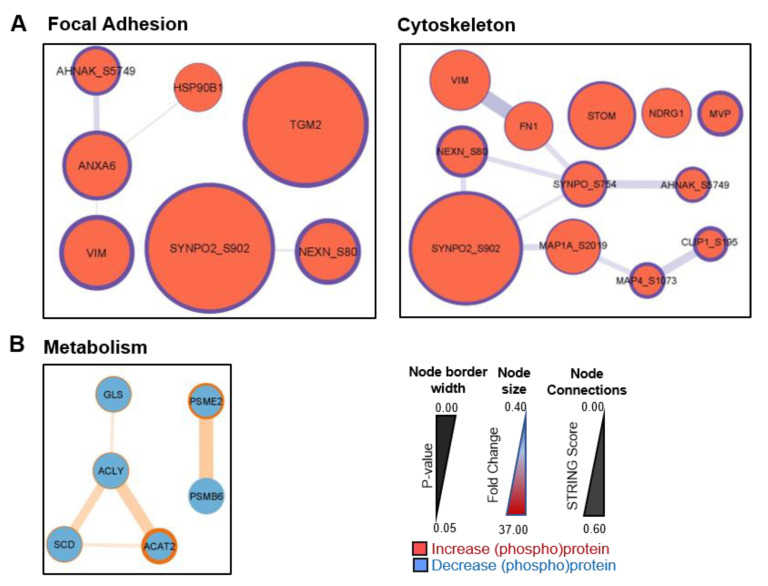
STRING/Cytoscape network analysis of differentially regulated BPH-1 proteins and phosphosites. Enriched signalling networks of (**A**) increased and (**B**) decreased BPH-1 (phospho)proteins in CAF co-culture. Node size is representative of protein expression fold change (co-culture/monoculture); node border width is representative of the *p*-value for co-culture/monoculture comparison for each protein; node connection line thickness is representative of STRING score between connected proteins.

**Figure 6 cancers-15-00699-f006:**
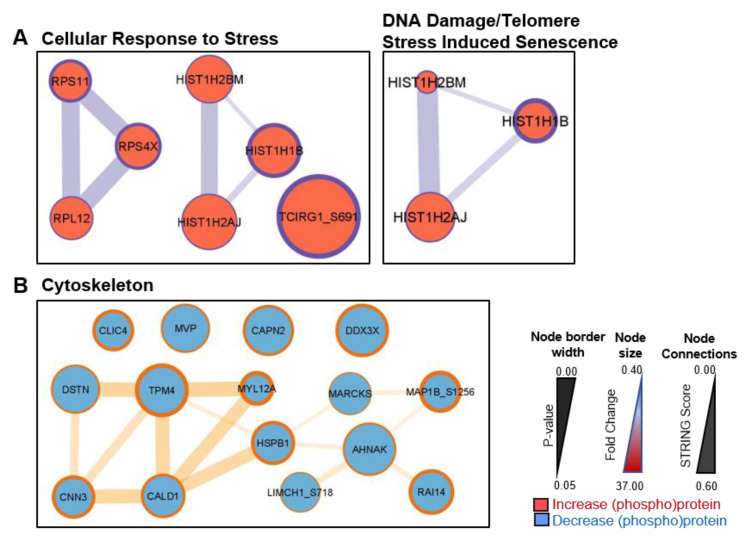
STRING/Cytoscape network analysis of differentially regulated CAF proteins and phosphosites. Enriched signalling networks of (**A**) increased and (**B**) decreased CAF (phospho)proteins in BPH-1 co-culture. Node size is representative of protein expression fold change (co-culture/monoculture); node border width is representative of the *p*-value for co-culture/monoculture comparison for each protein; node connection line thickness is representative of STRING score between connected proteins.

**Figure 7 cancers-15-00699-f007:**
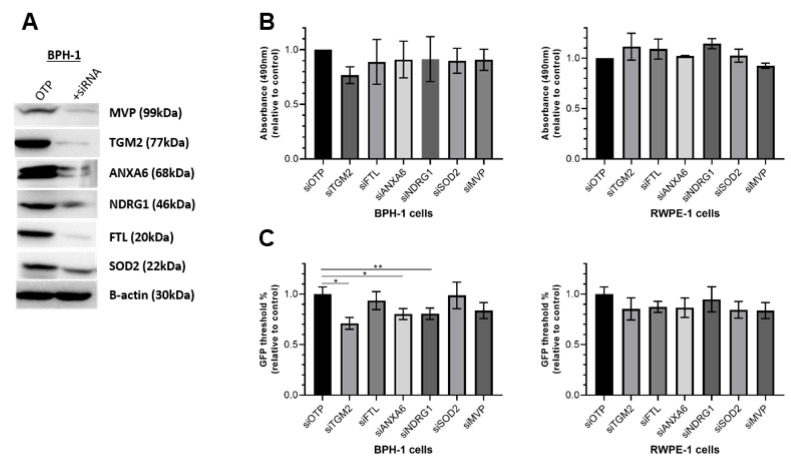
Functional interrogation of selected candidates via proliferation assays. (**A**) Representative Western blot images demonstrating siRNA-mediated knockdown. (**B**) MTS assays for BPH-1 and RWPE-1 benign prostate epithelial cells after 72 h of monoculture. Data are expressed relative to 0 h and normalized to the control (siOTP). (**C**) Epithelial cell proliferation in co-culture. GFP-positive BPH-1 and RWPE-1 cells were cultured with GFP-negative CAF cells for 72 h. GFP fluorescence intensity was used as a measure of epithelial cell number, relative to 0 h and normalized to the control. Mean ± SEM, *n* = 3. One-way ANOVA. * *p* < 0.05, ** *p* < 0.01.

**Figure 8 cancers-15-00699-f008:**
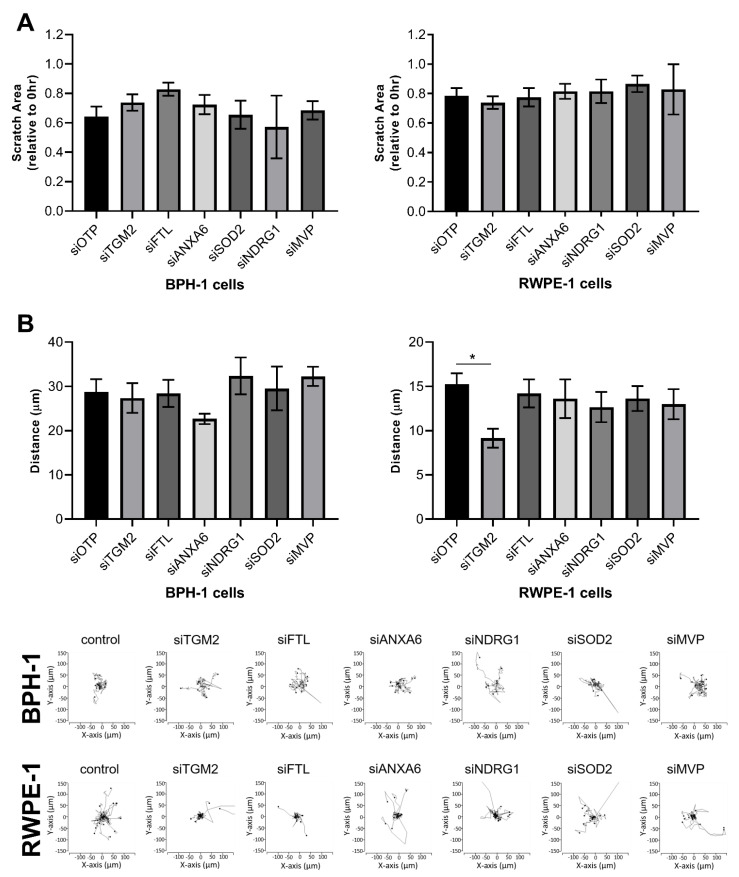
Functional interrogation of selected candidates via migration assays. (**A**) Scratch closure assay of BPH-1 and RWPE-1 benign prostate epithelial cells. Data represent scratch area 16 h post-scratch relative to 0 h timepoint. Mean ± SEM, *n* = 3. (**B**) Random cell migration assay of prostate epithelial cells in co-culture with CAFs. GFP-positive BPH-1 and RWPE-1 cells were cultured with GFP-negative CAF cells over 16 h. GFP-positive cells were tracked for total distance travelled. Mean ± SEM, *n* = 4. One-way ANOVA. * *p* < 0.05. Representative plots of *x*- and *y*-axis movements for BPH-1 and RWPE-1 cells in co-culture with CAFs are shown underneath.

**Table 1 cancers-15-00699-t001:** BPH-1 Protein fold-change in co-culture experiments. Top differentially regulated proteins in BPH-1 cells in CAF co-culture compared to monoculture (>twofold change in either direction and *p* < 0.05) compared to BPH1 in NPF co-culture vs. monoculture. Data are displayed as mean fold-change of raw protein intensity from mass spectrometry measurement ± SD from three biological replicates and *p*-value calculated from Student’s *t*-test comparing monoculture and co-culture. * *p* < 0.05, ** *p* < 0.01, *** *p* < 0.001.

BPH-1 Protein Fold-Change				
Protein	BPH-NPF/Monoculture		BPH-CAF/Monoculture	
	Fold-Change ± SD	*p*-Value	Fold-Change ± SD	*p*-Value
TGM2	23.51 ± 9.10	** 0.002	35.72 ± 10.27	** 0.003
FTL	22.60 ± 17.46	* 0.026	32.90 ± 12.73	*** 0.000
STOM	13.78 ± 10.71	0.057	26.10 ± 20.71	* 0.020
VIM	14.85 ± 5.32	*** 0.000	13.21 ± 3.16	*** 0.000
ANXA6	12.26 ± 4.57	* 0.012	15.64 ± 14.00	* 0.010
NDRG1	11.40 ± 16.33	0.135	15.69 ± 20.80	* 0.042
FN1	11.24 ± 5.28	0.081	10.06 ± 6.67	* 0.036
FAM114A1	6.19 ± 3.31	0.087	8.34 ± 5.01	* 0.027
TGFB1	4.89 ± 1.66	* 0.010	8.97 ± 6.43	* 0.033
MVP	5.99 ± 2.22	** 0.001	6.52 ± 2.00	*** 0.000
LGALS1	2.95 ± 1.78	0.154	4.34 ± 1.74	* 0.010
CTSB	1.77 ± 1.23	0.318	2.53 ± 0.91	** 0.005
ACAT2	0.47 ± 0.30	0.079	0.43 ± 0.12	** 0.004
PSME2	0.60 ± 0.11	* 0.045	0.47 ± 0.10	* 0.017
SCD	0.64 ± 0.04	0.216	0.48 ± 0.25	* 0.035
GLS	0.41 ± 0.37	0.153	0.50 ± 0.26	* 0.045

**Table 2 cancers-15-00699-t002:** Fibroblast protein fold-change in co-culture. Top differentially regulated proteins in CAF cells in BPH1 co-culture compared to monoculture (>twofold-change in either direction and *p* < 0.05) compared to NPF in BPH-1 co-culture vs. monoculture. Data are displayed as mean fold-change of raw protein intensity from mass spectrometry measurement ± SD from three biological replicates and *p*-value calculated from Student’s *t*-test comparing monoculture and co-culture. * *p* < 0.05, ** *p* < 0.01.

Fibroblast Protein Fold-Change				
Protein	NPF-BPH/Monoculture		CAF-BPH/Monoculture	
	Fold-Change ± SD	*p*-Value	Fold-Change ± SD	*p*-Value
SRSF7	1.92 ± 0.30	* 0.034	2.96 ± 1.08	** 0.008
PRKDC	2.79 ± 2.05	0.058	2.39 ± 0.96	* 0.015
HIST1H4A	1.72 ± 0.41	* 0.016	2.18 ± 0.29	* 0.010
HIST1H2AJ	2.34 ± 1.34	* 0.044	2.09 ± 0.35	* 0.038
GBA	2.22 ± 0.94	0.056	2.09 ± 0.53	* 0.011
MYL12A	0.27 ± 0.09	* 0.013	0.24 ± 0.08	** 0.001
RPS27A	0.52 ± 0.07	** 0.001	0.40 ± 0.05	** 0.002
TAGLN	0.85 ± 0.28	0.749	0.40 ± 0.06	* 0.037
CLIC4	0.50 ± 0.11	** 0.007	0.42 ± 0.08	* 0.011
RAB5A	2.00 ± 2.47	0.840	0.45 ± 0.15	* 0.046
MGST1	0.43 ± 0.12	0.095	0.46 ± 0.19	* 0.040
MARCKS	1.08 ± 0.79	0.841	0.48 ± 0.25	* 0.041
CNN3	0.44 ± 0.15	0.068	0.45 ± 0.05	** 0.007
HSPB1	0.44 ± 0.13	* 0.044	0.47 ± 0.09	** 0.007
CALD1	0.61 ± 0.11	0.058	0.48 ± 0.03	* 0.017
ITGAV	0.59 ± 0.23	0.241	0.51 ± 0.25	* 0.039

## Data Availability

The MS proteomics data were deposited in the ProteomeXchange Consortium via the PRIDE partner repository [50] with the dataset identifier PXD037272. Other raw data can be obtained from the corresponding author.

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
