# Peer review of "Cell-Type-Specific Signalling Networks Impacted by Prostate Epithelial-Stromal Intercellular Communication"

_cancers, 2023, doi:10.3390/cancers15030699_

Round 1
Reviewer 1 Report (Previous Reviewer 1)
Thank you for taking my comments into consideration and appropriately revising the manuscript. Added and revised figures/text have addressed my critique.
Author Response
Thank you for your comments
Reviewer 2 Report (Previous Reviewer 2)
The authors have addressed my comments appropriately.
Author Response
Thank you for your comments
Reviewer 3 Report (Previous Reviewer 4)
While I agree with all answers to my questions except the question of doing qPCR analysis to look for change in transcriptome vs proteome. It will be very interesting for the following two reasons.
1) It will provide a clear distinction whether the change happens at transcriptomic level or at proteomic level.
2) The experiment can be done in a non contact based co culture and will be an additional piece of data towards whether the intercellular communication is mediated through secreted soluble mediators or through direct cell cell interaction.
Round 2
Reviewer 3 Report (Previous Reviewer 4)
The concerns have been addressed professionally by authors.
This manuscript is a resubmission of an earlier submission. The following is a list of the peer review reports and author responses from that submission.
Round 1
Reviewer 1 Report
This manuscript has used cell type specific labelling with aminoacid precursors CTAP to characterize proteomic and phospho-proteomics changes in the crosstalk between prostate epithelial cells and normal/cancer fibroblasts. Study is well designed and data clearly presented. Here some minor queries for authors to address:
1. Experiments shown in Figures 2, 3, and 4 indicate that researchers used 3 experimental replicates. Are replicates biological, from different donors, or the same donor (technical replicates)? Please clarify, if different donors, it would be important to perform a PCA on proteomics of CAFs vs NPFs to infer if proteomics depend more on cancer phenotype or biological donor.
2. Figure 3, if there are only 3 replicates, please show all as single data points. These can be overlaid on the bar plots.
3. Figure 7 show some promising results in relation to migration phenotype. Please add representative images/movies as evidence to back up these results.
Reviewer 2 Report
The central theme of this work is the poorly understood interactions between cancer associated fibroblasts and adjacent epithelium, and understanding of which should suggest new therapeutic options. New techniques to examine these interactions are welcome. This is an elegant study whose importance lies as much in the technological developments as in the pathways elucidated. The ability to identify proteins produced in a specific cell type in a co-culture environment is a useful tool and the rapid improvements now taking place in proteomic technology suggest that further sensitivity improvements will be coming in the near future.
Protein based analyses are always more biologically convincing than transcriptomic approaches. However, the latter are more commonly used due to their simplicity and to the advances in technology that have made these approaches dominant in many areas. I wonder if the authors ran parallel studies on mRNA expression of the identified targets. If so do are the results congruent? I say this accepting the difficulties and limitations in separating mRNA signals in co-culture situations.
Limitations of the data provided are that the main output is broad pathway categories rather than defining specific molecular outputs that can be targeted. Another limitation, of course, is the simplicity of the system, and the potential need to include additional cell types such as leukocyte populations. However, those are considerations for the future.
Overall this is an interesting technical report that highlights some potentially interesting pathways.
Minor points
Line 53 – I assume the authors mean significant side effects or some synonym, rather than comorbidities (the simultaneous presence of two or more diseases)
Line 54/55 singular/plural dissonance “prostate cancer that are”
Line 135 – what does “early” mean. Could you specify, below 6 passages, or whatever. Also purists would argue that these are not cell “lines” but low passage cultures.
Line 148 The first use of the term “heavy” would seem to require some explanation (also “light”) the facts that these are deuterated crops up almost in passing on the third use of the term. I also can’t find a source for the heavy reagents.
Reviewer 3 Report
Questions:
1) “Amongst the different cell types that comprise the TME, the role of fibroblasts has received particular attention, as these cells are the most abundant and main component of the TME.” (raw 62-64) – need reference to research article confirming the last part of the sentence.
2)“Whole tissue was enzymatically digested to release cells into suspension” (124-125). It is necessary to clarify which enzymes were used for tissue digestion.
3)Which transfection agent was used for transfection BPH-1 cells with LyrM37-KDEL and CAF/NPF cells with DDCM.tub-KDEL?
4)Epithelial cells and fibroblasts were maintained at different atmosphere (normal and 5% O2, respectively). It is necessary to clarify of atmosphere content in co-culture experiments.
5) In section “2.7. Western Blot analysis”: It is necessary to specify secondary antibodies and primary antibodies for housekeeping protein (b-actin?).
6) Which method of transfection was used for gene knockdown by siRNA (section 2.7)?
7) Why were fibroblasts infected with a retrovirus with RFP? There isn’t explanation of it in manuscript. In co-culture proliferation assay, GFP fluorescence measurement was method for estimation of BPH-1 cells count. However, fluorescence spectrum of GFP and RFP can overlay. How authors did discriminated fluorescence signals from epithelial cells and fibroblasts?
8) There are not samples for control untransfected BPH-1, NPF and CAF cells on Figure 1B.
9) In the legend to figures 2 and 4 it is necessary to clearly write what is displayed on the axes, what units of measurement are indicated on them.
10) In the legend to figures 8A it is necessary to specify time point of assay (in hours).
Reviewer 4 Report
1. It would be interesting to see whether perturbation of upstream activators or downstream targets of TGM2 complement the observed phenotype.
2. Authors have shown through cell-type specific labelling with amino acid precursors (CTAP), changes in phospho proteome of prostate epithelial cells when co-cultured with normal patient-derived prostate fibroblasts (NPFs) versus matched CAFs. Authors should provide experimental evidence whether these changes were due to change in gene expression or through stabilization/degradation of proteins under these specific conditions. A qPCR analysis of the identified target genes and/or a pulse chase experiment would provide a critical piece of data.
3. Authors have not clearly mentioned whether these co-culture experiments were of contact or non-contact nature and the explanation there of are missing.
4. The research article has been neatly written with acronyms and abbreviations properly explained for greater accessibility.